

# Identifying Source Region Elemental Indicators in Aged Saharan Dust Plumes Over the Tropical Atlantic

Daniel E. Yeager[1] and Vernon R. Morris[1,2]

[1]NOAA Center for Atmospheric Sciences, Howard University, Washington, D.C., United States
[2]School of Mathematical and Natural Sciences, Arizona State University, Phoenix, AZ, United States

*Correspondence to*: Daniel Yeager (daniel.yeager7@gmail.com); Vernon Morris (Vernon.Morris@asu.edu)

**Abstract.** This work examines the spatial dependency of Saharan dust aerosol composition over the Tropical Atlantic Ocean using observations collected during the 2015 Aerosols and Ocean Science Expedition (AEROSE). Regionally specific

elemental indicators remain detectable in the dust samples collected along the Saharan air layer trajectory far into the Tropical Atlantic marine boundary layer. Saharan dust transport characteristics and elemental composition were determined by Inductively Coupled Plasma Mass Spectrometric (ICP-MS) analysis of airborne dust samples, ship-based radiometry, satellite aerosol retrievals, and atmospheric back-trajectory analysis. Three strong dust events (SDEs) and two trace dust events (TDEs) were detected during the campaign. The associated mineral dust arrived from potentially 7 different north African countries

within 5 to 15 days of emission, according to transport analysis. Peak Na/Al and Ca/Al ratios (>1 and >1.5, respectively) in dust samples were traced to northern Saharan source regions in Western Sahara and Libya. In contrast, peak Fe/Al ratios (0.4-0.8) were traced to surface sources in southern Saharan regions in central Mauritania. We observe the highest ratios of $\frac{Ca+Mg}{Fe}$ (3-10) at sampling latitudes north of 15N in the Atlantic. Additionally, the sub-micron fraction of dust particulate settling over the Atlantic showed significant temporal and spatial variability, with coarse-fine Al ratios (at 0.8 microns) of 1.05, 0.65, and

0.95 for SDE1 (11/21-23), SDE2 (11/25-26), and SDE3 (11/28), respectively. This was consistent with elemental concentrations of Ca, Na, K, Ti, and Sr, per Al, that exhibited coarser size tendencies per dust event. These observations could validate spatially-sensitive aerosol models by predicting dust aerosol abundance and composition within the tropical Atlantic. Such predictions are critical towards understanding Saharan dust effects on regional climate, Atlantic Ocean biogeochemistry, satellite observations, and air quality modeling.




# 1 Introduction

The Sahara Desert emits 600-1000 Tg of mineral dust particulate into the atmosphere per year (Kaufman et al., 2005; D'Almeida, 1986; Kok et al., 2021). These dust aerosols primarily originate from fluvial sediments formed in extinct waterways within the nearly 9-million km² desert (Prospero et al., 2002). These loose sediments are suspended by convergent
trade winds (regional Harmattan winds) and dry convection across North Africa. The resulting dust aerosols are transported within hot and dry air layers, collectively called the Saharan air layer (SAL), which cover an altitude of roughly 1-5 km (Ismail et al., 2010; Tsamalis et al., 2013). This SAL can transport dense dust outflows thousands of kilometers from their source regions (Washington et al., 2009). A substantial portion of these dust aerosols enter the Atlantic Ocean as the SAL overrides the cooler Atlantic marine boundary layer (Ginoux et al., 2012).

These large quantities of Saharan dust extending over the Atlantic have numerous environmental impacts on regional climate, health, ecosystems, and Earth observations from satellite. Suspended mineral dust directly affects climate radiative balance through particle scattering and absorption of solar and terrestrial energy (Miller et al., 2014). This direct effect also complicates satellite retrievals of surface and atmospheric parameters (e.g. sea surface temperatures, surface reflectance, and chlorophyll
concentrations) (Volpe et al., 2009; Luo et al., 2019; Maddy et al., 2012). Alternatively, indirect climate effects occur when dust alters cloud formations modulating Earth's radiative balance (Hoose and Möhler, 2012). As Saharan dust particulate settles into oceans and foreign soils, its iron and phosphorous content impacts ecosystem productivity and carbon cycling (Zhang et al., 2015). These settling dust particulates have also been linked to upper respiratory illnesses, especially within finer-sized particulate matter less than 2.5μm and 10μm (PM2.5 and PM10) (Prospero and Mayol-Bracero, 2013). The
magnitude of these environmental effects is highly dependent upon the dust aerosol size, shape, and composition.

Currently, there is a large uncertainty in Saharan dust aerosol composition due, in part, to limited in-situ observations, especially over the Atlantic Ocean. These silicate-based dust aerosols contain variable amounts of iron oxides, titanium oxides, carbonates, quartz, and other trace minerals largely dependent upon the Saharan source region (Journet et al., 2014; Formenti
et al., 2014). The mineral content changes throughout atmospheric transport due to gravitational settling, aerosol surface chemistry, and water uptake (Di Biagio et al., 2014; Pye, 1987). Operational models of climate, atmospheric chemistry, and radiative transfer have not fully resolved the space and time dependence of these aerosol characteristics. This has resulted in quasi-static assumptions of Saharan dust aerosol size, shape, composition, and resulting optical properties (Kok et al., 2017; Di Biagio et al., 2017). These dust aerosol modeling biases limit the prediction of Saharan dust climate and ecological effects.
Therefore, theoretical assumptions of aged Saharan dust aerosol composition need validation and advancement through in-situ characterizations.



## 1.1    Objective

Using ship-based field campaign observations, this work examines the spatial dependency of the aerosol composition of Saharan dust plumes extending over the tropical Atlantic. During the observing period, various Atlantic dust outflows were detected and compared by dust particulate composition, size dependency, and transport characteristics. The results of this investigation provide unique observations of the aging and evolution of Saharan dust aerosols crossing the Atlantic. We show that the composition of these Saharan dust outflows retains the distinguishing characteristics of their source regions during their transit over the tropical Atlantic. In the following section, we provide an overview of the ship campaign, onboard dust measurements, and relevant data analysis techniques.

## 2    Methods

### 2.1    AEROSE Campaign

The Aerosols and Ocean Science Expeditions (AEROSE) are a series of ship campaigns designed to characterize African aerosol evolution, understand Atlantic climate effects of African aerosol intrusions, and improve regional satellite observations and climate models (Morris et al., 2006; Nalli et al., 2011). AEROSE focuses its set of in-situ, radiometric, and vertical profiling measurements on the tropical Atlantic Ocean, where in-situ aerosol measurements are sparse and discontinuous. This sampling region is highly favorable for investigations of the atmospheric processing of mineral dust due to its proximity to Saharan and sub-Saharan aerosol outflows while also allowing favorable satellite calibration and validation opportunities (Nalli et al., 2018, 2006). This investigation focused on the 2015 AEROSE campaign aboard the NATO Research Vessel (RV) *Alliance* due to the high frequency and intensity of dust plume observations relative to other campaigns.

AEROSE collaborates with the Pilot Research Moored Array in the Tropical Atlantic-Northeast Extension (PNE) project. PNE is a joint effort between the United States National Oceanic and Atmospheric Administration (NOAA), France, and Brazil to maintain and expand moored buoys that retrieve critical measurements of oceanic and maritime meteorological conditions. These buoy measurements provide a rich supply of information for both weather and climate prediction while requiring routine visits for sensor maintenance (Bourlès et al., 2008).

The 2015 PNE/AEROSE campaign began aboard the NATO RV *Alliance* in the Canary Islands on November 15[th], with an initial southwestward path towards the first buoy located at approximately 20N, 38W, that was reached on November 20th, as shown in Figure 1.



85

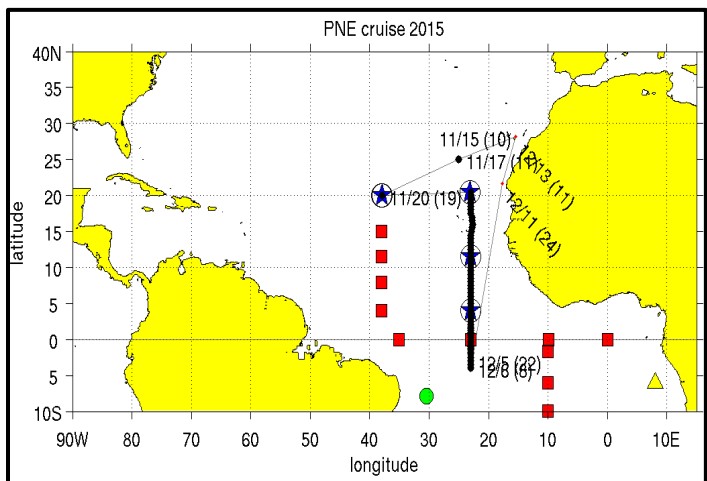

**Figure 1. Cruise path and Atlantic buoy sites for the 2015 AEROSE campaign (from NOAA/AOML website at https://www.aoml.noaa.gov/phod/pne/cruises.php). USA, France, and Brazil buoy sites are shown in blue stars, red squares, and green circles, respectively.**

90

After reaching the first buoy site on November 20th, the NATO *Alliance* headed eastward to the second buoy site at approximately 20N, 23W, that was reached on November 25th. The RV then headed southward along the 23W parallel from November 25th to December 6th, where the *Alliance* reached its southernmost point of the cruise at approximately 2S. Three more buoys, two US and one French, were transected along this southern leg of the cruise. An accelerated return to port was

95 made along the 23W parallel from December 6th to December 11th. From December 11th through December 13th, the cruise path turned northeastward back to port in the Canary Islands where the campaign concluded.

## 2.2 Overview

To satisfy research objectives, we utilize sun photometer, spectroradiometer, and ambient aerosol filter samples to detect and classify various Saharan dust outflows throughout AEROSE '15. If Al concentrations, a proxy for relative mass, exceeded

1ppm (according to filter sample analysis) during elevated sun photometer aerosol optical depth (AOD) retrievals (>0.3) or 90[th] percentile spectroradiometer dust retrievals, the sampling periods were considered Strong Dust Events (SDEs). The remaining sampling periods with detectable Al concentrations were classified as Trace Dust Events (TDEs).

To determine dust sample sources, we examined air parcel back-trajectory intersections with moving averages of Visible

Infrared Imaging Radiometer Suite (VIIRS) Level 3 (L3) satellite AOD retrievals. NOAA Hybrid Single Particle Lagrangian Integrated Trajectory (HYSPLIT) simulations were initialized in hourly increments, spanning 24 hours, from dusty AEROSE



observation points at 10m and 4km altitudes. Back-trajectory durations ranged from 96 to 240 hours, depending on endpoint proximities to Saharan source regions. VIIRS L3 AOD retrievals were averaged with spatial and temporal sensitivity to 5-day back-trajectories. This resulted in VIIRS AOD mosaics with moving 5-day zonal averages bounded by the endpoints of 5-day

HYSPLIT trajectories. These products allowed for comparisons of dust event sources, aerosol lifetimes, and transport pathways.

For each dust event, this study examines the concentration of Al, Ca, K, Na, Mg, Fe, Ti, and Sr elements within respective dust samples. A focus is placed on normalized elemental concentrations (non-Al element to Al ratio), (Ca+Mg)/Fe ratios, and

size-resolved Al-27 concentrations. These elemental ratios have shown source region sensitivity in previous Saharan dust aerosol field experiments (Formenti et al., 2011; Marconi et al., 2014; Scheuvens et al., 2013).

### 2.3    Ship Radiometry

For this study, the Marine Atmospheric Emitted Radiance Interferometer (M-AERI) was used for spectral detection of Saharan dust plumes throughout AEROSE '15. This sea-going Fourier transform infrared (FTIR) spectrometer measured high-

resolution infrared (IR) radiances from upward and downward viewing angles in nearly 5-minute intervals (Minnett et al., 2001). Based on principles discussed in (Nalli et al., 2006), we utilize two narrow spectral channels for dust detection, namely 961 cm$^{-1}$ and 1231 cm$^{-1}$. Both of these channels are relatively insensitive to water vapor and ozone, but the 961 cm$^{-1}$ has much higher relative sensitivity to mineral dust; thus, we can use the following simple equation for dust detection:

$$Dust\ Signal = \frac{T_B(961) - T_B(1231)}{T_B(961) + T_B(1231)}\ , \qquad (1)$$

where $T_B$ represents the wavenumber dependent brightness temperature. The resultant dust signal was resampled to 1-hour resolution. The M-AERI was operational during the campaign from 11/17/2015 through 12/06/2015.

Microtops handheld sunphotometers provided daytime, multi-channel, aerosol optical depth (AOD) measurements from the visible to near-infrared spectrum of atmospheric columns during sun-viewing opportunities. These measurements were a part

of NASA's Aerosol Robotic Network-Marine Aerosol Network (AERONET-MAN) observation system that provides satellite and aerosol transport model validation (Smirnov et al., 2009). Sunphotometer measurements were acquired in the 440, 500, 675, 870, and 936nm solar spectrum bands during AEROSE '15. Level 2.0 AOD series datasets were used throughout this investigation.

### 2.4    Aerosol Elemental Analysis

One-stage and Six-stage Staplex cascade impactor air samplers, shown in Figure 2, were used throughout the AEROSE '15 campaign for size dependent collection of aerosol particulate matter onto quartz fiber filters.


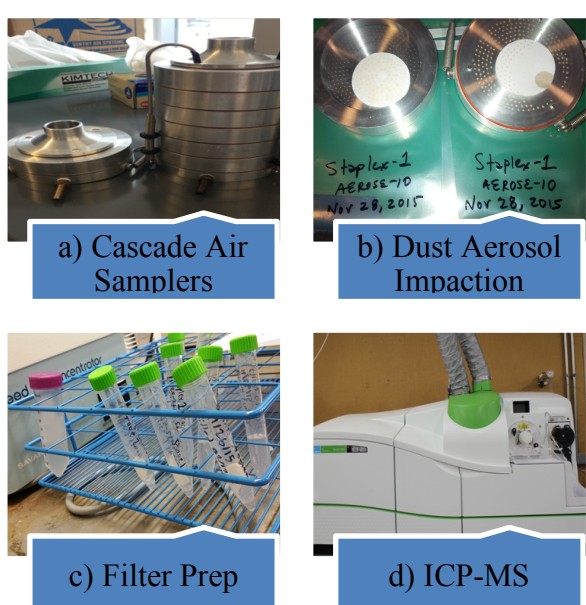

a) Cascade Air Samplers

b) Dust Aerosol Impaction

c) Filter Prep

d) ICP-MS

**Figure 2. a) One-stage and Six-stage Cascade air samplers deployed throughout AEROSE '15, b) dusty quartz fiber filter samples collected on 11/28/2015, c) quartz fiber filter quarters placed in nitric acid solution, d) ICP-MS used for trace metal analysis of dusty filter samples**

The Staplex cascade impactor uses a vacuum pump to draw in air flow through a series of stages that gradually decrease in nozzle diameters. Due to inertial flow, larger particles are incrementally inhibited from passing to lower stages. Aerosol particulate was collected on quartz fiber filters at each stage for subsequent chemical analysis. The two-stage air sampler allowed for size dependent cutoffs in aerosol collection at 0.8µm diameter, with Stage 1 allowing aerosol particles greater than 0.8µm diameter, and Stage 2 allowing aerosol particles with diameters less than 0.8µm. During the campaign, only the bottom three stages of the six-stage air sampler collected sufficient dust particulate for analysis. The bottom three stages consisted of aerosol diameter ranges of 1.1- 2.1µm (Stage 5), 0.65-1.1µm (Stage 6), and 0.65µm > (Stage 7). 10µm and 2.5µm pre-impactors were utilized throughout air sampling to discriminate against larger sea spray aerosols. One-stage and Six-Stage air samplers operated from 11/21/2015 through 12/12/2015. After sampling, filter samples were stored in air-tight plastic containers and analyzed for elemental composition.

Inductively Coupled Plasma Mass Spectrometry (ICP-MS) analysis detected trace metal compositions of AEROSE '15 air sample filters. Detectable isotopes related to Saharan dust particulate included Al (isotope: 27), Ca (isotopes: 42, 43, 44, 46, 48), K (isotope: 39), Na (isotope: 23), Mg (isotopes: 24, 25, 26), Fe (isotopes: 54, 56, 57, 58), Ti (isotopes: 46, 47, 48, 49, 50), and Sr (isotopes: 84, 86, 87, 88). Sample preparation for ICP-MS analysis involved placing approximately a quarter of each filter sample in 20% nitric acid solution (with 18 mega Ω distilled water) for at least 24 hours before extracting the resulting supernatant liquid for ICP-MS analysis. The supernatant liquid from the filter samples was introduced to the ICP-MS, where





the resulting plasma-induced ions could be separated and detected by the coupled mass spectrometer in terms of pulse intensity (measured in counts per second [cps]). In order to determine sample element concentrations, a standard with known elemental concentrations (PerkinElmer's Instrument Calibration Standard 2) was simultaneously introduced to the ICP-MS to create an intensity calibration curve (linear regression) that could be interpolated for elemental concentration. Calibration curves featured standard concentrations of 0.01, 0.05, 0.1, 0.5, and 1 ppm of each elemental isotope of interest to this investigation

and were verified for linear correlations ($R^2 > 0.9$).

## 3    Results and Discussion

### 3.1    AEROSE Dust Event Classifications

M-AERI dust signals, which ranged from roughly 0.3-0.4, reached the 90[th] percentile during the 11/20-11/22, 11/26, and 11/27 timeframes, and as high as the 80[th] percentile after 11/27. Spatially, these dust signals intensified near the (20N, 35W), (18N,

23W), (13N, 23W) coordinates, as shown in Figure 3a. From 11/19 to 12/05, sun photometer measurements recorded AOD peak values on 11/20 (0.33), 11/22 (0.51) and 11/28 (0.45), while the remaining measurement dates recorded AOD < 0.3. A similar range of peak AOD values (0.3-0.5) was observed on 12/09 as the Alliance RV returned northward to port along the 23W parallel between 10 and 15N (not shown in Figure 3a). Al concentrations in filter samples exceeded 1ppm during the 11/21-11/26, 11/28, and 12/03 collection dates (Figure 3b). Conversely, only trace amounts of Al (1ppm >) were observed on

11/27 and 12/09. Based on these observations, we classify filter samples collected on 11/21-11/23, 11/25-11/26, 11/27, 11/28, and 12/09 as SDE1, SDE2, TDE1, SDE3, and TDE2, respectively.

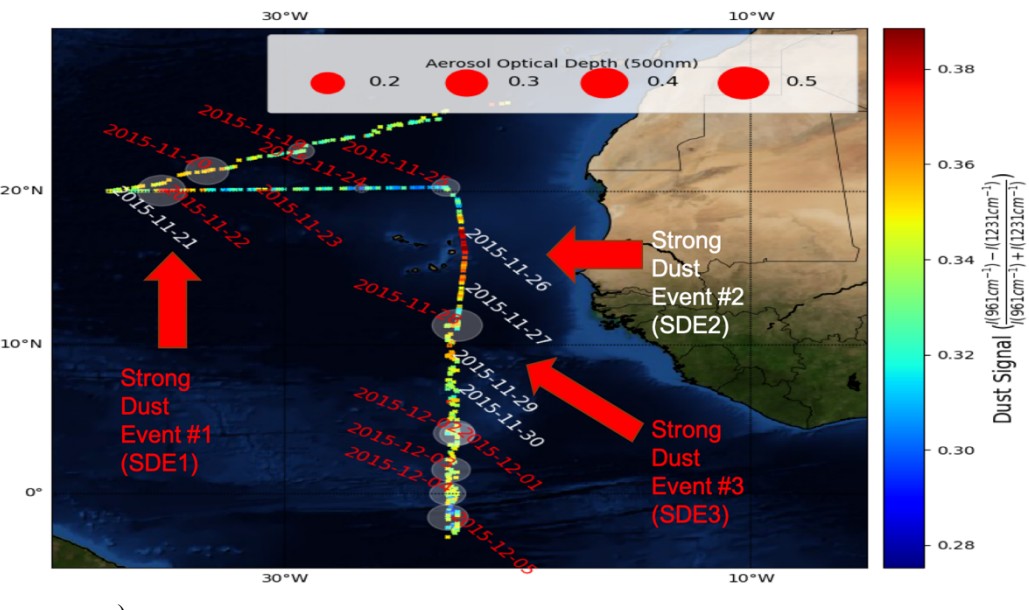

a)


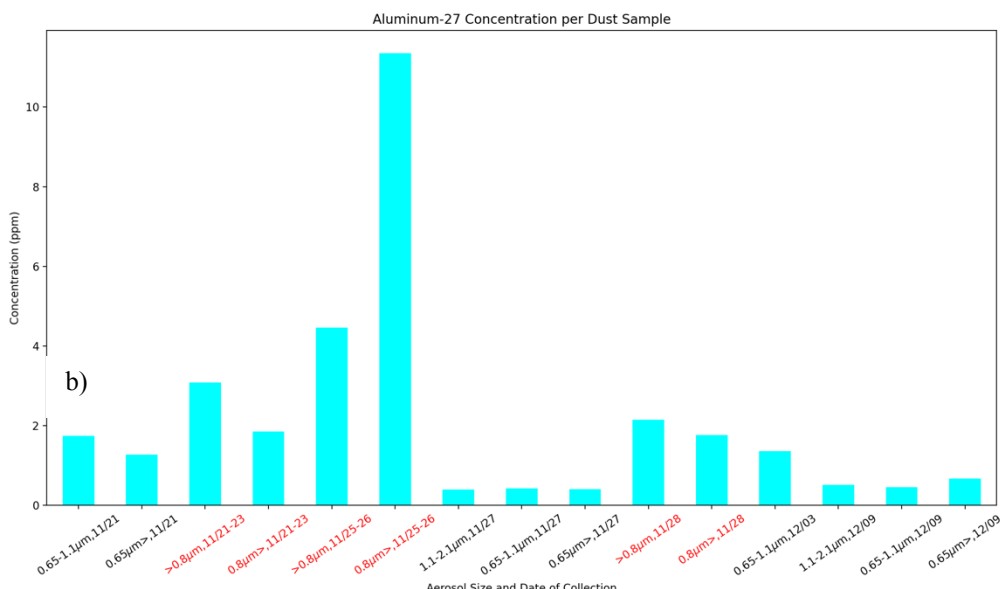

**Figure 3. (a) Sun photometer AOD values at 500 nm (transparent filled circles) and M-AERI dust signal observations defined by Eq. (1) (color-coded circles) collocated to AEROSE ship locations from 11/19 to 12/05. Dates shown in red represent sun photometer measurement days, while white represents non-measurement days due to cloud coverage. (b) Filter Al-27 concentrations (ppm) per date and particle size during AEROSE '15. Red x-labels indicate comparable heavy dust event samples.**

When comparing SDE1-SDE3, SDE2 had the largest accumulation of fine (0.8μm > D) dust aerosol particulate. Two-Stage Staplex coarse-fine ratios (at 0.8 microns) for Al-27 were 1.05, 0.65, and 0.95 for SDE1 (11/21-23), SDE2 (11/25-26), and SDE3 (11/28), respectively.

To summarize radiometric and filter-based Al measurements, five distinctive dust air masses (originating from three SDEs and two TDEs) were detected during the 2015 AEROSE sampling period. Air sample filters validated remotely sensed aerosol values from the M-AERI, sun photometer, and subsequent satellite retrievals. We find that the abundance of settling dust aerosol particulate is proportional to the sun photometer and M-AERI dust signals. When examining SDE1-SDE3 Al concentrations, we find that the reduction in Al concentration in SDE3 is consistent with the shift from 90[th] to 80[th] percentile peak M-AERI dust signals. Furthermore, the elevated surface Al concentrations (> 1 ppm) from SDE1 to SDE3 concur with coincident sun photometer AOD retrievals in the range of 0.3-0.5. In terms of aerosol size, the size-resolved Al ratios suggest substantial temporal and spatial variability exists within the sub-micron fraction of dust particulate settling into the Atlantic. This was evident by the shift in Al concentration from the fine to coarse particle size thresholds (0.8 μm cutoff) during SDE2. Overall, the large dust signal, comparable diameters of aerosol collections, and differing collection sites of SDE1-SDE3 and



TDE1-TDE2 supported spatial dependency studies of Saharan dust composition during AEROSE '15. In the following sections, we compare the transport and composition of these dust events.

## 3.2 Dust Event Transport Analysis

### 3.2.1 SDE1 Transport

Based on transport analysis in Figure 4, SDE1 particulate was linked to dust emission in central Sahara (Chad and Niger) and northwestern Sahara (Western Sahara and N. Mauritania). The bulk of SDE1 particulate would have likely been transported from its closest and farthest source regions, Western Sahara and Bodélé Depression, within 10 and 15 days, respectively.

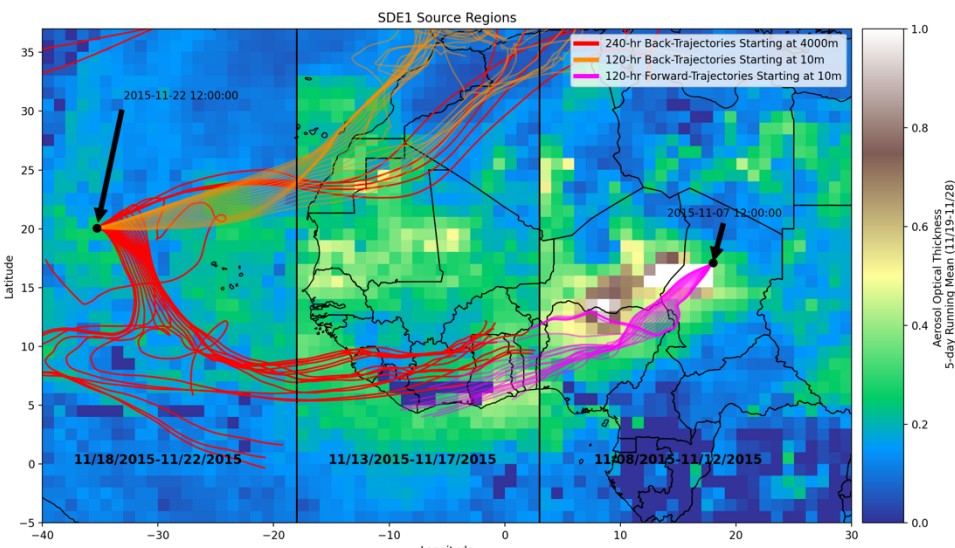

**Figure 4. 5-day VIIRS AOD mosaic for three sequential swaths (11/17-22, 11/12-17, 11/07-11/12), and NOAA HYSPLIT trajectories from ship (orange back-trajectories begin at 10m and red trajectories begin at 4km) and Bodélé Depression (magenta forward-trajectories begin at 10m).**

Within the 11/17-11/22 zonal average, VIIRS AOD ranged between 0.3 and 0.4 near the AEROSE DE1 sampling location on 11/22. In the adjacent 11/12-11/17 swath, AOD values between 0.4 and 0.5 overlaid much of Western Sahara, Mauritania, and western Mali. Large swaths of AOD values between 0.3-0.4 were also visible just offshore in the Gulf of Guinea. In the easternmost 11/07-11/12 swath, AOD values greater than 0.7 overlaid western Chad, southern Niger, and northern Nigeria during the 11/07-11/12 timeframe

SDE1 10-day back-trajectories, starting at 10m altitudes, terminated east and northeast of SDE1 observations, with intersections in high AOD (0.4-0.5) portions of Mauritania and Western Sahara. The majority of 10-day back-trajectories, starting at 4km, terminated in non-Saharan countries bordering the Gulf of Guinea (Togo, Benin, Ghana). To verify the





potential sources of these trajectories, we simulated 5-day forward trajectories from the nearby and active Bodélé Depression. These forward trajectories intersected high AOD values in Chad, southern Niger, Northern Nigeria, and 10-day (4km altitude) back-trajectories from the ship.

### 3.2.2 SDE2 Transport

Based on transport analysis from Figure 5, SDE2 particulate was closely linked to dust emission in Western Sahara. Most of this dust particulate would have likely arrived within 5 days of source region emission.

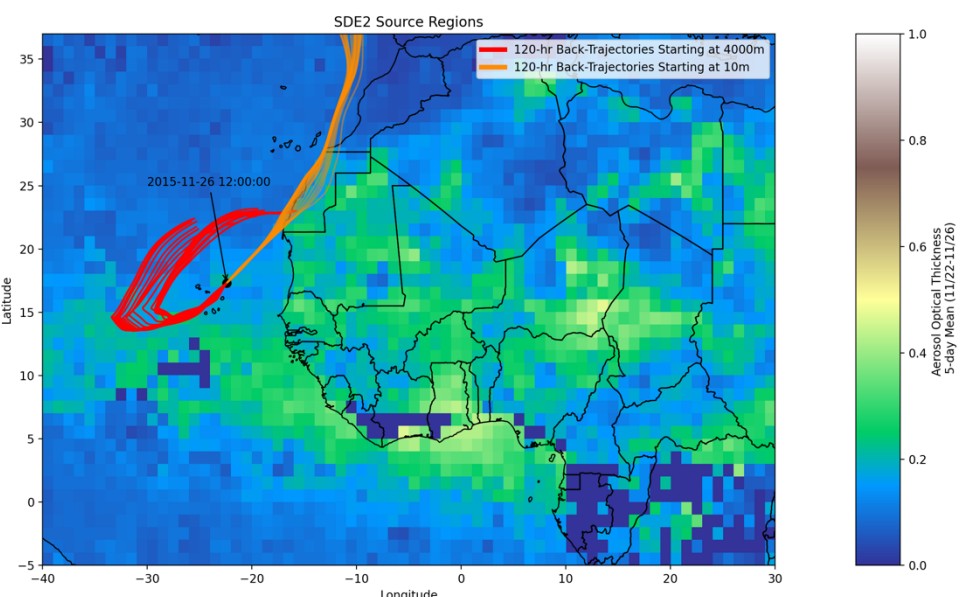

**Figure 5. Averaged VIIRS AOD from 11/23 to 11/26 and 120hr NOAA HYSPLIT back-trajectories (orange back-trajectories begin at 10m and red trajectories begin at 4km) from ship location on 11/26.**

Within the single 11/23-11/26 AOD mosaic, a small swath of VIIRS L3 AOD between 0.3 and 0.4 overlaid much of the sampling site on 11/26. AOD values between 0.3 and 0.4 overlaid most of Western Sahara, Mauritania, and Mali. In comparison to Figure 4, central Saharan AOD values decreased to a 0.3-0.5 range.

The majority of 5-day ship back-trajectories, beginning at 10m, intersected Western Sahara (AOD values 0.3-0.5). 5-day back-trajectories, beginning at 4km, show a clockwise trajectory toward termination points just offshore of Western Sahara.

### 3.2.3 TDE1 Transport

According to transport analysis in Figure 6, TDE1 particulate was closely linked to dust emission in Mauritania and Western Sahara. Most of this dust particulate would have likely arrived within 5 days of source region emission.





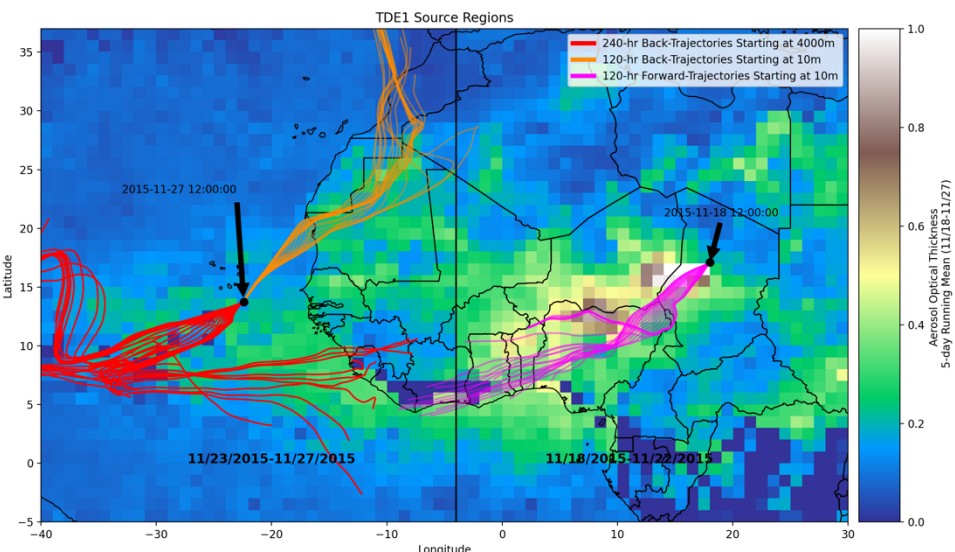

**Figure 6. 5-day VIIRS AOD mosaic for two sequential swaths (11/23-27 and 11/18-22), NOAA HYSPLIT back-trajectories from ship (orange back-trajectories begin at 10m and red trajectories begin at 4km), forward trajectories from the Bodélé Depression (pink trajectories begin on 11/18 at a 10m altitude).**

A swath of VIIRS L3 AOD values between 0.2 and 0.4 coincided with ship-based observations of SDE2 on 11/27. AOD values between 0.2 and 0.4 overlaid most of Western Sahara, Mauritania, and Mali. In comparison to Figure 5, central Saharan AOD values > 0.5 expanded across central Chad and Niger. Spatial domains of AOD > 0.3 decreased over much of Western Sahara in comparison to SDE2 observations from Figure 5.

5-day back-trajectories, beginning at 10m, intersected optically thick sites (0.3-0.4) in Mauritania and Western Sahara. While 9 of 24 4km back-trajectories terminated east of the TDE1 sampling site, all trajectories remained west of Ivory Coast. Thus, there is low confidence in substantial dust sample contributions from Central Saharan origins.

### 3.2.4    SDE3 Transport

SDE3 particulate was closely linked with dust emissions across parts of central Mauritania, northern Mali, southwestern Algeria, Chad, and Niger, as shown in Figure 7. SDE3 dust particulate would have likely been transported from its farthest source regions within 10 days.



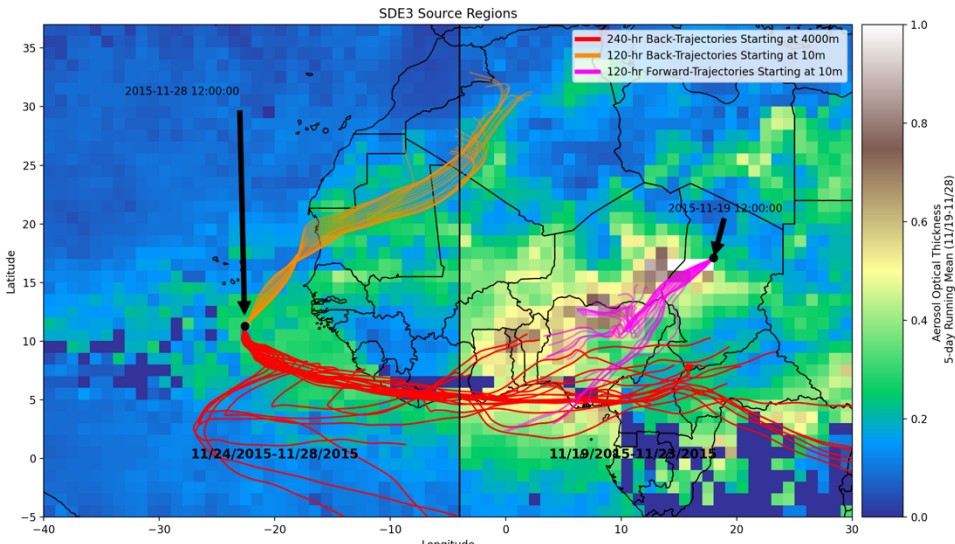

**Figure 7. 5-day VIIRS AOD mosaic for two sequential swaths (11/23-28 and 11/19-11/23), NOAA HYSPLIT back-trajectories from ship (orange back-trajectories begin at 10m and red trajectories begin at 4km), forward trajectories from the Bodélé Depression (pink trajectories begin on 11/19 at a 10m altitude).**

During the 11/23-11/28 timeframe, a large swath of VIIRS AOD values near 0.4 coincided with ship-based observations of SDE2. Pockets of AOD between 0.2 and 0.4 covered small amounts of Western Sahara, a majority of Mauritania, and southwest Algeria. In contrast, AOD > 0.6 spanned across SW Chad, eastern Niger, and the central Sahel during the 11/19-11/23 timeframe.

5-day back-trajectories, beginning at 10m, intersected much of central Mauritania, northern Mali, and SW Algeria. These trajectories intersected AOD values between 0.2 and 0.4 as they crossed Mauritania and southwest Algeria. 10-day back-trajectories, beginning at 4km, traveled eastward along the coastline of the Gulf of Guinea. As these back-trajectories passed east of -5W, they intersected large swaths of AOD values > 0.5 within the Sahel. 5-day forward-trajectories, beginning at 10m altitudes within the Bodélé Depression, terminated near the eastern Gulf of Guinea. These trajectories intersected high AOD values in Chad, southern Niger, and northern Nigeria, as well as 10-day back-trajectories from the Alliance.

### 3.2.5 TDE2 Transport

TDE2 dust particulate was linked to emission from eastern Libya, Bodélé Depression, Niger, southwestern Algeria, Mali, and southern Mauritania (shown in Figure 8). Dust sample particulates were likely transported from their farthest source region within 10 days. Within the sampling location, high VIIRS AOD values contrasted relatively low Al concentrations in filter samples. While this may be associated with higher ship speed during the sampling period, it may also indicate that substantial dust particulate remained aloft.




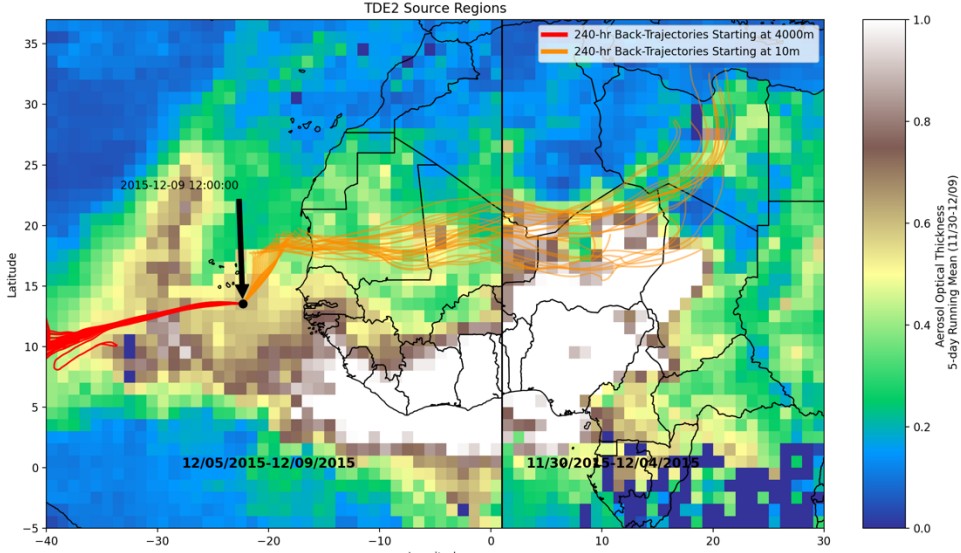

**Figure 8. 5-day VIIRS AOD mosaic for two sequential swaths (12/05-12/09 and 11/30-12/04), and NOAA HYSPLIT back-trajectories from ship (orange back-trajectories begin at 10m and red trajectories begin at 4km)**

Within the 12/05-12/09 zonal average, a large swath of AOD > 0.5 coincided with ship-based observations of SDE2. This large swath covered much of the latitudes between 5N and 15N. AOD between 0.3 and 0.5 covered most of Mauritania, Western Sahara, and Mali. In the 11/30-12/04 zonal average, AOD > 0.8 overlaid much of western Chad, Niger, and West African countries bordering the Gulf of Guinea. A relative increase in AOD (0.4-0.6) was also observed in Northeastern Libya.

10-day ship back-trajectories, beginning at 10m, intersected much of central Mauritania, Mali, Niger, northern Chad, and Libya. These back-trajectories intersected large swaths of AOD > 0.4 in each of these countries. Contrastingly, 10-day back-trajectories, beginning at 4km, traveled westward of the ship and away from potential Saharan source regions.

### 3.3    Linking Dust Event Composition to Source Regions

Normalized elemental concentrations varied substantially per dust event and sample size. Na, Ca, and Fe elements exhibited

00    the largest variability, with standard deviations of 0.33, 0.28, and 0.20, respectively. Composition variability, shown in Figure 9, was attributable to Saharan source region activity discussed during dust event transport analysis.

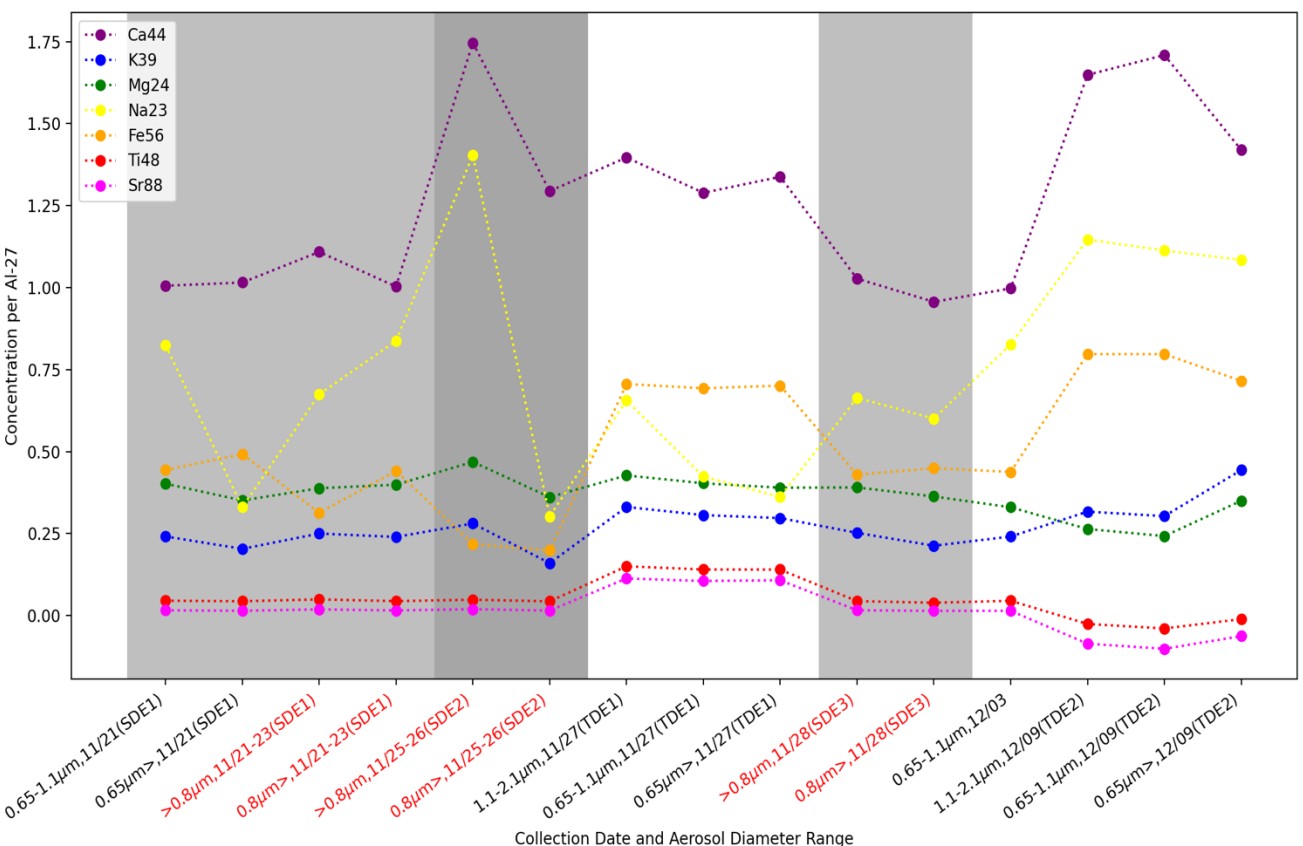

**Figure 9. Elemental ratios of dusty air sample filters collected throughout the AEROSE '15 campaign. The x-axis represents the date, particle size, and dust event designation of each dust sample. Ca-44 (purple), K-39 (blue), Mg-24 (green), Na-23 (yellow), Fe-56 (orange), Ti-48 (red), and Sr-88 (magenta) concentrations were normalized by Al-27 concentrations for each filter sample.**

### 3.3.1 Calcium

Dust sample Ca/Al ratios ranged between 1 and 1.8 throughout the AEROSE '15 campaign. Ratios exceeded 1.5 during SDE2 and TDE1 when particle sizes exceeded 0.8 and 0.65 microns, respectively. The lowest ratios (near 1) occurred during SDE1 and SDE3, whereas intermediate ratios (near 1.3) occurred during TDE1. Ca/Al ratios were highest in the coarsest size fraction for all dust events except TDE2 (slight peak in the 0.65-1.1μm range).

The peak in Ca/Al during SDE2 validated transport analysis that identified Ca-rich, Western Saharan soils as the primary source region. The abundance of Ca in northwestern Saharan soils was consistent with previous dust sample observations (Rodríguez et al., 2020; Journet et al., 2014; Kandler et al., 2007). Comparably high Ca ratios during TDE2 were linked to Libyan sources also shown to be Ca-rich; (Marconi et al., 2014) observed Ca/Al ratios of $1.9 \pm 0.7$ for northern Libyan dust aerosol samples. The coarser size tendency of Ca/Al per dust event matches findings from Kandler et al. 2007 and suggests





greater size sensitivity is necessary for modeling the distribution of calcium-bearing dust minerals. When examining SDE1, SDE2, and TDE1, each linked to Ca-rich Western Sahara, we find that the Ca/Al ratios were proportional to the proximity to Western Sahara. This is consistent with the size tendencies observed since presumably larger Ca-bearing particulate would be lost disproportionately as a function of aging.

### 3.3.2 Iron

Fe/Al ratios ranged between 0.2 and 0.8 during the AEROSE '15 campaign. We observed the highest Fe/Al ratios, 0.7-0.8, during the trace measurement periods of TDE1 and TDE2. Size-averaged Fe/Al ratios were 0.4, 0.2, and 0.4 for SDE1, SDE2, and SDE3, respectively. When comparing SDE1-SDE3 and TDE1-TDE2, there were no consistent size dependencies for Fe/Al ratios throughout the AEROSE '15 campaign.

Fe/Al increased as potential source regions expanded southward into arid Sahel regions. These regions included central Mauritania, Mali, Niger, and Chad, which were shown to be higher in iron content (Formenti et al., 2014; Di Biagio et al., 2017; Journet et al., 2014). The highest Fe/Al ratios (0.4-0.8) during TDE1, SDE3 (highest SDE), and TDE2 shared central Mauritania back-trajectory intersections, which agrees well with Saharan soil mapping efforts from (Journet et al., 2014). In contrast, the lowest Fe/Al ratio, during SDE2, coincided with highly exclusive emissions from Western Sahara. Notably, previous studies have grouped Western Sahara and Mauritania into one potential source area (PSA2) (Scheuvens et al., 2013), but our observations of Ca and Fe from these respective countries supports the need for more spatially distinctive PSA classifications. We do observe a finer size tendency amongst the highly aged dust particulate (15 days >) during SDE1 One-Stage and Six-Stage air samples. However, this observation was not validated with comparably aged dust events during the AEROSE '15 campaign.

### 3.3.3 Sodium

Na/Al showed the highest variability across dust events and sampling size. Peak Na/Al ratios occurred within the D>0.8um and 1.1-2.1um size fraction of SDE2 and TDE2, respectively. Na/Al ratios ranged from 0.6 to 0.8 for SDE1 and SDE3. We observe Na/Al minimums (0.3-0.4) within the finest fractions of Six-Stage SDE1 samples and One-Stage SDE2 samples. With the exception of SDE1, Na/Al ratios were highest in the coarsest fractions for all dust events.

Western Sahara, the most likely SDE2 source region, has an abundance of sebkhas producing precipitated Na-sulphate, Na–carbonate, and NaCl evaporites, which could explain the high Na/Al values (Rodríguez et al., 2011). Similarly, northern Libya has been shown to have an abundance of sebkhas (Yan et al., 2016), which may explain the similarly high Na/Al ratios during TDE2.





### 3.3.4    Magnesium

Mg/Al ratios showed smoother variances with an absolute range of 0.2-0.5 throughout AEROSE '15. The absolute maximum and minimum occurred within the coarse fraction of SDE2 and the 0.65-1.1um fraction of TDE2, respectively.

The sole peak of Mg/Al during SDE2 and transport analysis suggests greater amounts of dolomite and magnesium carbonate minerals were transported from Western Saharan events.

### 3.3.5    Potassium

K/Al also showed smooth variances with absolute minimums and maximums observed within the fine fraction of SDE2 and the D<0.65µm fraction of TDE2. With the exception of TDE2, K/Al ratios showed coarse fraction size tendencies during air sampling.

K/Al increases occurred during transition periods, which favored central Mauritania, Mali, and Algeria sources. The abundance of K-bearing illite minerals within the clay fraction may explain the relative increases in K-Al during TDE1 and TDE2 (Journet et al., 2014).

### 3.3.6    Titanium and Strontium

Ti/Al and Sr/Al showed the smoothest variance and had strongly correlated ratios. Maximums (near .1) occurred during the transition event (TDE1) between SDE2 and SDE3. Both elements showed consistent coarse fraction size dependencies for each dust event.

We find that Ti/Al and Sr/Al ratios during TDE1 were 20 to 200 times greater than those observed by (Rodríguez et al., 2020), where dust sample back-trajectories remained mostly above 20N. AEROSE Ti/Al and Sr/Al peak ratios coincided with back-trajectories over inner-Morocco.

### 3.3.7    Sampling Latitude Dependency of Dust Composition

Elemental ratios of Ca, Mg, and Fe concentrations showed distinctive sampling latitudinal dependencies, as shown in Figure 10.



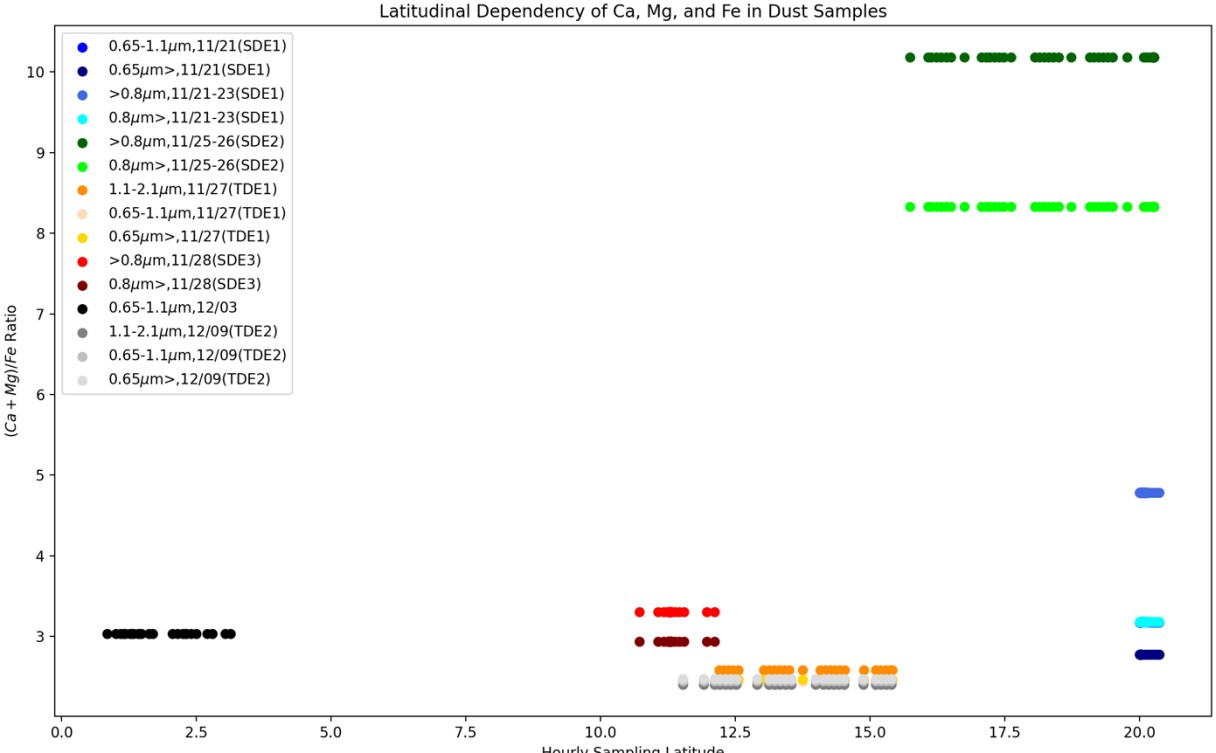

**Figure 10. (Ca+Mg)/Fe ratios as a function of AEROSE '15 hourly sampling latitude. Blues (11/21-11/23; SDE1), greens (11/25-11/26; SDE2), oranges (11/27; TDE1), reds (11/28; SDE3), black (12/03), and greys (12/09; TDE2) represent various sampling dates.**

The highest ratio and range, roughly 3-10, occurred between 15N and 20N, while ratios < 3 occurred between 12 and 16 N, and ratios of 3-3.5 occurred below 12 N. The distinction in Atlantic dust sample (Ca+Mg)/Fe ratios above (3-10) and below (2-4) 15N resembles the latitudinal dependencies of soil mineralogy across the Sahara Desert (Scheuvens et al., 2013). Transport analysis showed that as dust source regions shifted from exclusively Northern Saharan, SDE2, to mixtures of Northern and Southern Saharan, (Ca+Mg)/Fe ratios decreased to a range of 2-5. The coarser size fractions generally exhibited higher (Ca+Mg)/Fe ratios than the finer fractions (except for TDE2), indicating that as dust aerosol size increased, Fe concentrations decreased and Ca concentrations increased. SDE1 ratios (blues) were the only dust collections not along the 23W parallel (farther west), which may also explain the lesser (Ca+Mg)/Fe ratios. These ratios suggest that aged dust particulate may lose calcium carbonate minerals at a greater rate than iron oxide minerals as coarser size fractions diminish with transport.

## 4    Conclusions

This work examined the spatial dependency of Saharan dust composition in the Tropical Atlantic using AEROSE '15 ship observations. The elemental variability observed in dust aerosol samples showed a dependency on source region, atmospheric





lifetime, and particle size, as shown in Table 1. These unique observations of Saharan dust aerosol evolution indicate
that its composition remains dynamic well into the Tropical Atlantic marine boundary layer.

| Dust Event | Dust Aerosol Age | Source Region(s) | Composition | Particle Size Indicators | Optical Properties |
|---|---|---|---|---|---|
| SDE1 (11/20-11/22) | 15 days > | Bodélé Depression, Western Sahara, and Northern Mauritania | Iron- and calcium-depleted (Fe/Al: 0.38; Ca/Al: 1.1) | Al Coarse-Fine Ratio: 1.05; | Peak $AOD_{500nm}$: 0.5; $\alpha = 0.1$ LWIR Dust Signal: 90th percentile |
| SDE2 (11/26) | 5 days > | Western Sahara | Iron-depleted (Fe/Al: 0.21), calcium-rich (Ca/Al: 1.52) | Al Coarse-Fine Ratio: 0.65; | LWIR Dust Signal: 90th percentile |
| TDE1 (11/27) | 5 days > | Mauritania and Western Sahara | Iron-rich (Fe/Al: 0.70), calcium-rich (Ca/Al: 1.34) | Dominant Fraction: 0.65-1.1 | LWIR Dust Signal: 90th percentile |
| SDE3 (11/28) | 10 days > | Central Mauritania, Northern Mali, and Southwest Algeria | Iron-rich (Fe/Al: 0.44); calcium-depleted (Ca/Al: 1.0) | Al Coarse-Fine Ratio: 0.95 | $AOD_{500nm}$: 0.5; $\alpha = 0.3$ LWIR Dust Signal: 80th percentile |
| TDE2 (12/09) | 10 days > | Central Mauritania, Central Mali, Bodélé Depression, and Northeast Libya | Iron-rich (Fe/Al: 0.77), calcium-rich (Ca/Al: 1.60) | Dominant Fraction: 0.65 um > | $AOD_{500nm}$: 0.4 $\alpha = 0.4$ |

**Table 1. Elemental, optical, and spatial characteristics of dust plume observations during the 2015 AEROSE campaign.**

This study combined elemental and geospatial analysis techniques to compare the origins and characteristics of Saharan dust
aerosol composition in the Tropical Atlantic. Based on AEROSE '15 dust sample composition, we find the utilization of
spatiotemporally dynamic satellite retrievals and aerosol dispersion models to be an effective means of deducing Atlantic dust
aerosol origin and lifetime. Aerosol transport assessments were also supported by the size-dependency of dust aerosol
collection and corresponding elemental analysis.

We identify iron and calcium as strong source region indicators in aged Saharan dust particulate crossing the Tropical Atlantic. The origins of iron-rich dust aerosol samples ($0.7 \geq$ Fe/Al) were concentrated in Sahelian countries, frequently including central Mauritania. In contrast, Ca-rich dust samples (Ca/Al $\geq$ 1.5) were most frequently observed at sampling and source region latitudes above 15N and 20N, respectively. In the future, these elemental indicators could validate source region emission and deposition contributions esti=mated by global model simulations. More specifically, these datasets could assess the feasibility of various African soil databases used to initialize Saharan dust model simulations. Additionally, the spatially dependent Fe deposition rates we observe may have significant implications on Atlantic Ocean carbon cycle estimates related to biogeochemical processes.

This study shows that elemental size dependency exists in Al, Ca, Na, K, Ti, and Sr elements composing Saharan dust aerosols in the Tropical Atlantic. We find that the coarser size dependency of Al (coarse-fine ratios 1.05) and Ca concentrations (1.5 in SDE2) reflects the shortest aerosol lifetimes observed throughout AEROSE '15. These observations could be indicative of how long source region elemental signals remain detectable in aged Saharan dust events. Such information could validate Saharan dust aerosol size and source region constraints prescribed by global dust aerosol models.

Our Atlantic observations of Saharan dust aerosol composition support a broader research initiative to incorporate spatiotemporally dynamic aerosol optical properties into environmental predictions. The heterogeneity of Atlantic dust aerosols we observe could have significant implications on the sign and magnitude of dust-climate radiative forcing assessments. Similarly, this work supports the need for more dynamic aerosol optical data to improve satellite retrievals of dust-laden atmospheres over the tropical Atlantic. We anticipate that future research will incorporate AEROSE dust aerosol datasets into existing observational frameworks for broader environmental modeling assessments.

**Code and Data Availability**

All code and data can be made available upon request by contacting DY and VM.

**Author Contributions**

DY designed the study, conducted the data analysis, wrote the draft of the paper, and interpreted the results. VM led the aerosol data collection efforts, provided feedback and data interpretation, as well as advisement on the paper.



**Acknowledgements**

This work was supported by the NOAA Cooperative Science Center in Atmospheric Sciences and Meteorology (NCAS-M) under the NOAA Educational Partnership Program Cooperative Agreement #NA16SEC4810006. We would like to thank the AEROSE and PNE science team members for their continuous environmental data collection efforts at sea. Dust elemental analysis was performed at the University of Maryland Baltimore County (UMBC) Molecular Characterization and Analysis Complex (MCAC). We acknowledge the NOAA Air Resources Laboratory for HYSPLIT transport and dispersion model access. Satellite products were acquired from the Level-1 and Atmosphere Archive & Distribution System (LAADS) Distributed Active Archive Center (DAAC), located at the Goddard Space Flight Center in Greenbelt, Maryland (https://ladsweb.nascom.nasa.gov/). We thank Dr. Nicholas Nalli for his technical review of the manuscript, NRV Alliance cruise data, and AERONET-MAN data contributions.

**Financial Support**

This work was funded by the NOAA Cooperative Science Center in Atmospheric Sciences and Meteorology (NCAS-M) under the NOAA Educational Partnership Program Cooperative Agreement #NA16SEC4810006

**Competing Interests**

The authors declare that they have no conflict of interest.

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
