# Peer review of "Identifying Source Region Elemental Indicators in Aged Saharan Dust Plumes Over the Tropical Atlantic"

_Atmospheric Chemistry and Physics, 2021_

## Referee Comment (RC1)

The manuscript submitted to the ACP journal by Yeager and Morris explores the spatial distributions of aerosol chemical composition in different grain-sized fractions over the North Atlantic Ocean during the AEROSE cruise 2015. Besides aerosol collection, the dust load in the air column was also determined using aerosol optical depth (AOD) retrievals using satellite and ship-based radiometry. The Saharan dust parcels from source regions in North Africa were traced using NOAA-HYSPLIT 5-day back trajectories coupled with AOD retrievals. The research topic of validating the role of atmospheric processing of mineral dust aerosols on its key elemental signatures during the cross-Atlantic Saharan dust transport is very crucial for the pre-instrumental Saharan dust provenance studies. However, the manuscript is not well written with vague objectives and methodology with major analytical flaws as given below:

(1) The substrate "quartz filter" used for aerosol collection is well known to have high metal blank levels. The sample collection on Whatman 41 (cellulose filter) is recommended for geochemical characterization of marine aerosols (Morton et al., 2013, published in the Journal of Limnology and Oceanography-Methods).

(2) The analytical protocol used is 24 hrs leaching of dust on  $1/4^{\text{th}}$  of the filter in 20 % HNO3, which does not guarantee 100% dissolution of aluminosilicates in mineral dust unless hydrofluoric acid is added. Before sample digestion, seasalt contribution should be removed by washing samples with ultrapure water (resistivity >18.2 MOhm-cm).

(3) No detail was provided on the ICPMS measurement protocol except for the target isotopes and concentrations of stock solutions. Major elements such Ca and Fe etc. can not be measured directly using ICPMS at standard resolution due to major isobaric and polyatomic interferences.

(4) The elemental abundances in partly leached dust fractions that likely contain seasalt and contributions from high total procedural blank do not make any sense.